# Effects of clown visits on stress and mood in children and adolescents in psychiatric care— Protocol for a pilot study

**Martina Zemp**[1,2]*, **Amos-Silvio Friedrich**[1☯], **Lorena Holzmeier**[1☯], **Simone Seebacher**[3], **Maggie Rössler**[4], **Urs M. Nater**[1,2]

**1** Department of Clinical and Health Psychology, University of Vienna, Vienna, Austria, **2** Research Platform "The Stress of Life – Processes and Mechanisms underlying Everyday Life Stress", University of Vienna, Vienna, Austria, **3** Department of Research and Learning, RED NOSES Clowndoctors, Vienna, Austria, **4** Department of Research and Learning, RED NOSES International, Vienna, Austria

☯ These authors contributed equally to this work.
* martina.zemp@univie.ac.at

**Funding:** RED NOSES Clowndoctors Austria provided financial support including travel expenses for the examiners and research materials (i.e., Salicaps collection devices). The IBL

## Abstract

Scientific evidence has shown that healthcare clowning can decrease the level of stress and anxiety in pediatric patients. However, little attention has been devoted to the potentially beneficial impact of clown visits in the child and adolescent psychiatry setting. Therefore, this pilot study aims at investigating short-term effects of clown visits by RED NOSES Clowndoctors Austria on stress and mood levels of children and adolescents in psychiatric care. The sample will consist of approximately 50 children and adolescents in inpatient psychiatric wards who receive clown visits on a weekly basis over four consecutive weeks. The examined intervention, i.e., the clown visits, is an integral part within the selected psychiatric institutions. Using a non-controlled pre-/post-test design, the level of salivary cortisol and self-reported stress and mood will be measured as primary outcomes before and immediately after each clown visit. Additionally, self-reported effects on care staff at the health care facilities will be assessed based on a questionnaire after each clown visit within the same time period of four weeks. Secondary outcome measures (i.e., health-related quality of life, emotional and conduct problems, perceived chronic stress) will be assessed at baseline and at close-out assessment after the four intervention weeks. Further control variables and potential moderators are included at baseline. Due to the nested data structure, multilevel modeling will be used to analyze the data. To our knowledge, this is the first study to examine the stress-reducing and mood-improving effects of clown visits on inpatients in child and adolescent psychiatry. Results will be relevant for the design of future large-scale RCTs and might provide valuable implications for the use of healthcare clowning to reduce stress and improve mood in children and adolescents in psychiatric care. The study is registered at ClinicalTrials.gov (Identifier: NCT04844398).

International GmbH, a Tecan Group Company, sponsored the test kits for the saliva analyses. The funders did not have any additional role in the study design, data collection and analysis, decision to publish, or preparation of the manuscript.

**Competing interests:** We confirm that there are no competing interests which might influence our work. Two authors are employed by the non-profit association RED NOSES Clowndoctors Austria [SS] and the foundation RED NOSES International [MR]. These authors primarily support the facilitation and administration of the study, but will not be involved in the decisions on the trial design, data collection and analysis, and decision to publish. This does not alter our adherence to PLOS ONE policies on sharing data and materials. The study was designed and conceptualized and is being conducted by the research team exclusively [MZ, AF, LH, UMN]. The specific roles of all authors are articulated in the 'author contributions' section.

# Background

Hospitalized children often face a range of demanding experiences. Stress and anxiety are prevalent phenomena during hospital stays [1, 2]. Preoperative anxiety in pediatric patients is one of the primary outcomes investigated in previous research [3–5], although children report other problems as well; for example, hospitalized children often feel "alone, [. . .] mad, and sad" [6]. Pain, loss of control, and disconnection from family and relatives can lead to physical and psychological impairments [7]. Even after discharge, children may exhibit symptoms of distress, such as restlessness and lower self-esteem [8]. In children in psychiatric care, stress and anxiety can be even more pronounced, as they might be part of the symptoms requiring inpatient treatment in the first place, and the process of adaptation to the context of hospitalization can exacerbate primary symptoms [9].

To address these issues and alleviate their impact on the patients' well-being, humor interventions have been identified and used as an effective treatment in hospitals for decades [10, 11]. Most commonly, these take the form of professional hospital clowns and are either carried out in independent visits or integrated into hospital routines. In Austria, such interactive clown performances are offered, inter alia, by RED NOSES Clowndoctors Austria, which is part of RED NOSES International (RNI), a non-profit association providing professional clown programs in the medical and social care settings. RNI identifies three target areas of healthcare clowning for clinical patients stemming from common problems associated with hospitalization: (a) Social disconnection: patients may be separated from family and friends for a prolonged period of time, resulting in feelings of loneliness and exclusion; (b) Painful emotions: patients may experience heightened levels of stress, fear, and anxiety; (c) Challenging environments: hospital environments may be unable to provide adequate psychosocial support, leading to feelings of boredom and powerlessness [12]. Healthcare clowning aims at addressing these issues by offering individual attention to patients in need, switching focus away from negative emotions to humorous play, and valuing a patient's unique strengths in an otherwise possibly impersonal environment. These encounters lead to improved communication, attention, energy, and mood [13]. Moreover, healthcare clowning has been conceptualized as a model for a more self-compassionate way of dealing with one's own shortcomings and hardships [14]. Presumably, a particularly suitable target group for the clowning approach are children and adolescents, who might respond especially well to clowning due to the immediate, non-intellectual approach [14]. Clown interventions in hospital settings are therefore assumed to be highly effective among pediatric inpatients.

Previous research on the effectiveness of healthcare clowning has mainly focused on interactive clown performances for children and adolescents in intensive care or undergoing medical surgery. Specifically, clown visits have been shown to reduce preoperative anxiety [15, 16] and stress [17], salivary cortisol [18], pain during medical procedures [19, 20], and the need for sedation [21] in pediatric patients. With the exception of a few pilot studies with adult patients [22–24], little attention has been paid to potential benefits of clowns in the psychiatric setting. For adult patients in a closed psychiatric ward, an interactive clown performance during a period of bi-weekly clown visits led to reduced disruptive behaviors, including less agitation, aggression, and self-injury [24]. To our knowledge, there is no prior study that has investigated the impact of clown visits in the setting of child and adolescent psychiatry. Hence, the current pilot study aims at addressing this gap.

A small number of studies have demonstrated positive effects not only for patients, but for medical staff as well. Anesthetists appreciated the presence of clowns accompanying their medical rounds [25], and professionals performing venipuncture benefitted from less anxiously agitated patients when assisted by clowns [26]. Most prominently, strong effects were

observed on nursing staff, where clown interventions have been shown to improve nurse-patient interactions [27, 28] as well as professional communication throughout their workday [28]. Ghaffari et al. [29] found that humor enhanced nurses' mental and emotional well-being; the authors further highlighted its ability to promote an energetic and creative state for nurses in clinical settings as well as help them to overcome sadness and despair [30]. This finding was supported by other studies reporting reduced negative mood in nurses experiencing the clown performances [28]. In a pilot study, Wild et al. [31] described pediatric psychiatric care staff reports about the helpfulness of healthcare clowning in their daily routine and their expressed support to continue the program—a finding that other scholars have reported similarly [28, 32].

## Study aims and hypotheses

To our knowledge, this pilot study is the first to investigate the impact of clown visits on stress and mood of children and adolescents in inpatient psychiatric care, examining patients' salivary cortisol in addition to self- and staff-reports on stress and mood perception. A second novelty is that we assess effects on stress and mood over the course of multiple visits, allowing for insight into dose-response relationships. We expect that children and adolescents involved in an interaction with the clowns will experience a decrease in their subjective and physiological stress levels as well as improved mood by redirecting their attention to the current pleasurable moment. Specifically, we hypothesize that (1) children and adolescents will report less perceived stress and improved mood and that (2) they will display a reduced cortisol level after the experience of a clown visit compared to before. We further expect that (3) the more frequently children and adolescents experience the weekly clown visits over the course of the four-week intervention, the stronger the stress-reducing and mood-improving effects will be. Last, we assume that (4) perceptions of care staff will indicate positive effects of the clown visits on their own individual mood, the atmosphere within the staff team, and the patients' well-being.

## Materials and methods

### Participants

This pilot study will be conducted with children and adolescents of any gender and psychiatric diagnoses who currently are inpatients in Austrian child and adolescent psychiatric institutions cooperating with RED NOSES Clowndoctors Austria. Specific inclusion criteria are (1) being aged between 7 and 18 years and (2) regular participation in clown visits at the relevant health care facility on a weekly basis. Exclusion criteria are (1) potential negative impacts of clown visits or study participation on participants' mood or well-being according to medical or paramedical care staff of the relevant health care facility and (2) insufficient command of German (for self-administered questionnaires only).

Participants will be recruited from two cooperating psychiatric healthcare facilities in Austria. The visits will take place in at least two different wards of each participating psychiatric care facility. In each ward, a maximum of 10 participants will be targeted. The four-week intervention assessments will take place twice in both clinics with different patients (time interval of approximately four to seven weeks between the two assessment phases). The first recruitment phase is initiated by a staff member of RED NOSES Clowndoctors, who regularly visits those clinics. Within the clinics, the recruitment process will be assisted by the care staff of the relevant wards. After being informed about the study goals and data collection process, care staff will evaluate the eligibility of specific patients regarding potential negative impacts of clown visits or study participation. If there are no anticipated negative consequences on

participants' health or well-being, informed consent will be obtained from participants and, in case of underage participants, their parents or legal guardians prior to the start of data collection.

Based on effect sizes reported in reference studies [18, 33–39] we computed the minimum sample size needed for identifying the expected effects assuming rudimentary paired one-tailed t-tests using G*Power [40]. Calculating with $\beta = .20$, $\alpha = .05$, and expected effect sizes for cortisol levels ($f = .25$), self-administered measures of stress ($f = .44$), anxiety ($f = .47$), and emotional state ($f = .23$), a minimum sample size of about 30 participants in total (children and adolescents together) is needed. It is of note that the effect sizes for emotional state were derived from adult samples due to a lack of child and adolescent data. All other estimates were based on child samples. Accounting for dropout and random variation, a sample size of at least 50 children and adolescents will be targeted.

## Trial design

This pilot study uses a non-controlled pre-/post-test design. There is no control group and no variation in intervention to be evaluated; thus, the study contains only one arm. See Fig 1 for an overview of the study schedule according to the SPIRIT 2013 Statement [41].

## Procedure

At baseline, standardized questionnaires will be used to assess participants' sociodemographic data, general mental and physical health status, emotional and conduct problems, perceived stress, and coping mechanisms. Participants will be administered paper-pencil versions of the undermentioned measures and will be asked to fill them out by themselves under supervision of the examiners. Additionally, the assigned care staff will provide basic clinical information for each participant in form of a self-administered questionnaire. In this session, participants will be assigned a pseudonymized code. Following baseline assessment, salivary cortisol as well as self-administered questionnaires on stress and mood states of participants will be collected before and after each clown visit on a weekly basis over four consecutive weeks. Immediately after the fourth clown visit, the self-administered questionnaires from the baseline assessing secondary outcome measures will be presented a second time. Effects on care staff at the health care facilities will be assessed within the same time period of four consecutive weeks based on a self-administered questionnaire completed after each clown visit. Fig 2 outlines the study procedure and assessments per time point.

## Intervention

Clown visits will be carried out routinely by two professional clown artists from RED NOSES Clowndoctors Austria once a week for four consecutive weeks either in a group setting or an individual setting. The selection of artists, type of setting (group or individual), duration of clown visits, and specific artistic sequences will correspond to usual clown visits in those institutions, following internally organized routines instead of being standardized. The essence of the clown visit is to catch the patient's attention proactively and reach the highest possible level of engagement. The specific artistic sequences and type of setting are implemented spontaneously according to the situational atmosphere and current condition of participants. One clown visit in the group setting will last between 1 and 2 hours including individual as well as collective attention depending on age, number of participants, and situational atmosphere. In the individual setting, each participant will be engaged individually between 5 and 20 minutes by the clown, depending on the situational mood and condition of the patients. The clown artists are not involved in any study-related research activities.

| | STUDY PERIOD | | | | | | | | | | |
| --- | --- | --- | --- | --- | --- | --- | --- | --- | --- | --- | --- |
| | Enrol-ment | Alloca-tion | Post-allocation | | | | | | | | Close-out |
| TIMEPOINT | $-t_1$ | 0 | $t_{1pre}$ | $t_{1post}$ | $t_{2pre}$ | $t_{2post}$ | $t_{3pre}$ | $t_{3post}$ | $t_{4pre}$ | $t_{4post}$ | $t_5$ |
| **ENROLMENT:** | | | | | | | | | | | |
| **Eligibility screen** | X | | | | | | | | | | |
| **Informed consent** | | X | | | | | | | | | |
| **Allocation** | | X | | | | | | | | | |
| **INTERVENTIONS:** | | | | | | | | | | | |
| *Clown visits* | | | ←———————————————————————→ | | | | | | | | |
| **ASSESSMENTS:** | | | | | | | | | | | |
| *Clinical data of participants reported by care staff (i.e., diagnoses/reasons for treatment, entry date to the clinic, number of previous clinic stays, current treatment, current medication, hospital school)* | X | | | | | | | | | | |
| *Sociodemographic data of children and adolescents (i.e., age, gender, type of school, first language, number of clown visits before enrolment), health-related quality of life, emotional and conduct problems, perceived chronic stress, coping with stress* | | X | | | | | | | | | |
| **Perceived stress, perceived mood, salivary cortisol** | | | X | X | X | X | X | X | X | X | |
| **Perceived enjoyment of clown visits** | | | | X | | X | | X | | X | |
| *Sociodemographic data of care staff (i.e., type of care staff, age, gender, extent and duration of employment in the relevant clinic, first language), Evaluation of clown visits by care staff (own mood, atmosphere within the care team, patients' well-being)* | | | | X | | X | | X | | X | |
| **Health-related quality of life, Emotional and conduct problems, Perceived chronic stress** | | X | | | | | | | | | X |

**Fig 1. SPIRIT schedule.**

## Primary outcome measures

**Perceived stress.**   Subjective stress levels of children and adolescents will be assessed in a self-administered manner with one item ("Right now I am feeling stressed.") using a 100-mm

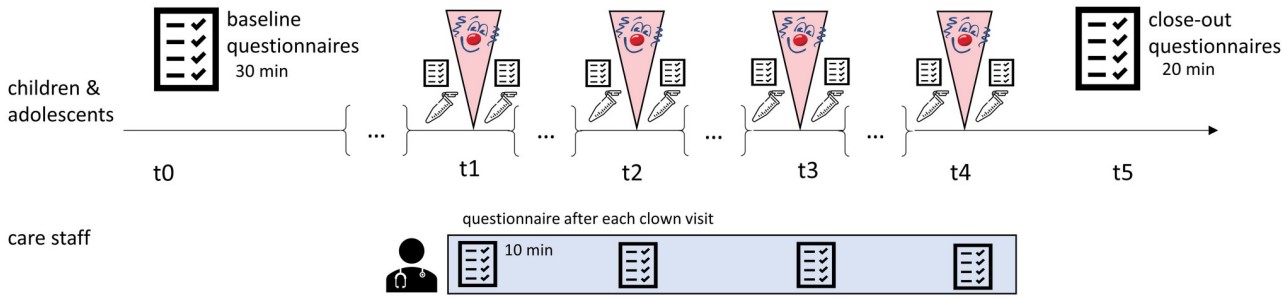

**Fig 2. Study procedure.**

visual analogue scale (VAS) before as well as immediately after each of the four clown visits. The item asks participating children and adolescents to draw a line where they would currently place themselves on the VAS ranging from 0 (not at all stressed) to 100 (very much stressed). Age-appropriate anchor graphics are placed at both extremes to support comprehension. The VAS approach has been used and validated in a number of previous studies in stress research [42, 43].

**Perceived mood.** The original German short version of the self-administered Multidimensional Mood Questionnaire (MDMQ) [44] will be used to measure adolescents' different dimensions of current mood states (pleasant–unpleasant; awake–sleepy; calm–restless) before as well as immediately after each of the four clown visits. The MDMQ provides four-item scales to measure each dimension. Participants respond to the statement "At this moment I feel. . ." by means of adjectives (e.g., "content"; "tired"; "agitated"; "well") on a 5-point Likert scale from 1 (not at all) to 5 (very much). The validity and reliability of the measure have been shown [45]. For children aged between 7 and 11 years, we adapted the scale to an age-appropriate picture-based version with two items per dimension rated on a similar 5-point Likert scale from 1 (not at all) to 5 (very much).

**Salivary cortisol.** Saliva samples of cortisol are collected as a marker for the physiological stress response of children and adolescents. Salivary cortisol reflects the activity of the HPA axis. We will use Salicaps collection devices consisting of collection tubes and straws. Samples will be collected immediately before as well as immediately after each of the four clown visits in a self-administered way, but under the supervision of the examiners. Participants will be thoroughly instructed to collect accumulated saliva for two minutes without swallowing. After the two minutes, participants will transfer the accumulated saliva into the Salicap tube via the straw. Tubes are then stored locally at the health care facility at -20°C prior to transport to and analysis in the biochemical laboratory.

### Secondary outcome measures

**Health-related quality of life.** Children's and adolescents' health-related quality of life will be assessed at baseline and at close-out assessment immediately after the fourth clown visit by the self-administered German revised questionnaire Health-Related Quality of Life (KINDL-R) [46, 47]. We will use the three subscales *physical health* (sample items: "During the past week I felt ill" or ". . . I had a headache or tummy-ache"), *mental health* (sample items: ". . . I had fun and laughed a lot" or ". . . I was bored"), and *self-esteem* (sample items: ". . . I was proud of myself" or ". . . I felt pleased with myself"). Each subscale consists of four items rated on a 5-point Likert scale from 1 (never) to 5 (always). Age-appropriate versions of the

questionnaire are used for children and adolescents. The validity and reliability of the German self-administered version have been demonstrated [48].

**Emotional and conduct problems.**   Children and adolescents will complete the five-item emotional problems and conduct problems subscales of the self-administered Strengths and Difficulties Questionnaire (SDQ) [49] adapted for use in German [50] at baseline and at close-out assessment immediately after the fourth clown visit. Sample items assessing *emotional problems* are: "I am often unhappy, depressed, or tearful" or "I have many fears; I am easily scared". Sample items measuring *conduct problems* are: "I take things that are not mine (from home, school, or elsewhere)" or "I get very angry and often lose my temper." Responses are 0 = not true, 1 = somewhat true, and 2 = certainly true. The validity and reliability of the German self-administered version have been supported in past research [51, 52].

**Perceived chronic stress.**   The self-administered Perceived Stress Scale (PSS) [53] assesses one's own subjective stress in the last month. Participants will be asked to evaluate the ten items on how often they experienced certain states, e.g., "In the last month, how often have you been angered because of things that were out of your control?" or "In the last month, how often have you found that you could not cope with all the things you had to do?" on a 5-point Likert scale ranging from 0 (never) to 4 (very often), once at baseline and at close-out assessment immediately after the fourth clown visit. A global value of perceived stress is obtained from summing up all responses after reversing positive items. The German version of the PSS will be used [54], which has been proven valid and reliable within a German sample including adults and adolescents [54]. Only adolescents from the age of 11 years will complete this questionnaire, as it is not age-appropriate for younger children.

**Evaluation of care staff.**   A self-administered questionnaire for care staff at the health care facilities assesses effects of clown visits on their own mood, the atmosphere within the care team, and the patients' well-being after each of the four clown visits. The questionnaire was developed by the authors specifically for the study purpose on the basis of previously implemented research tools [32, 55]. The items were partly self-developed and partly translated into German, adopted according to the study's objectives, newly designed and formulated in accordance with previous research [32, 55]. The questionnaire includes 20 items. Using a 4-point Likert scale from 1 (very negative) to 4 (very positive) with an additional fifth option (no change perceived), assigned care staff will assess the extent of individual changes according to specific emotion-related domains like energy level, concentration, sense of sympathy, or serenity. Following the same scaling, care staff is further asked to evaluate the impact of the clown visits on work related items like general atmosphere in the ward, teamwork within the care staff, or ability to provide empathic care to the patients.

## Potential control variables and moderators

**Control variables (examiners).**   The examiners that will carry out data collection procedures on-site will complete a self-administered questionnaire assessing the date of assessment, name of examiner, name of the health care facility, type of setting (group or individual), number of participants (if group setting), pseudonymized identification codes of participating children and adolescents, start and end time of the clown visit (duration), time of saliva sample collection (pre and post), and special incidents for each clown visit.

**Sociodemographic and clinical data and control variables for salivary cortisol (children and adolescents).**   A self-administered questionnaire to assess children's and adolescents' sociodemographic data includes information about gender, age, type of school attended before psychiatric admission, first language, and number of previously experienced clown visits. Further, psychiatric diagnoses or reasons for treatment, admission date at the clinic, number of

previous clinic stays, current treatment (forms of therapy), current medication, attending in-house clinic schooling, and somatic comorbidities are assessed by a self-administered questionnaire filled out by care staff. Control variables for salivary cortisol are measured by a self-administered questionnaire completed by children and adolescents (i.e., intake of food, chewing gums, or relevant drinks (i.e., coffee, tea, cola, juices, and alcohol); tooth brushing or smoking within an hour before the collection of the saliva samples; whether they did sport in the last 48 hours; whether they took medication on the same day; and whether they feel healthy).

**Sociodemographic data (care staff).** A self-administered questionnaire to assess care staff's own sociodemographic data includes information about type of care staff, age, gender, extent and duration of employment in the relevant clinic, and first language.

**Perceived enjoyment.** A self-developed and age-appropriate self-administered 5-point Likert scale with one item assesses how participants and care staff enjoyed the clown visit from 0 (not at all) to 4 (very much) after each of the four clown visits.

**Coping with stress.** The self-administered Stress and Coping Questionnaire for Children and Adolescents (SSKJ 3–8) [56] is a German measure to assess stress and coping styles in children and adolescents. At baseline, participating children and adolescents will complete the coping style scale of the SSKJ 3–8, encompassing 30 items that describe coping strategies to reduce stress. Participants are prompted in written form to evaluate how regularly they exhibit the described behaviors on a 5-point Likert scale ranging from 1 (never) to 5 (always). These correspond to five subscales consisting of six items each, namely *seeking social support* (e.g., "I tell one of my family members what happened"), *problem-focused coping* (e.g., "I decide for a way to solve the problem"), *avoidance* (e.g., "I act as if everything was o.k."), *constructive-palliative coping* (e.g., "I try to do something relaxing"), and *destructive-anger-focused coping* (e.g., "I get angry and destroy something"). The SSKJ has shown good validity (convergent, discriminant) and reliability (Cronbach's alpha, retest) for all subscales [57].

## Statistical analysis

Due to the nested data structure (outcomes being nested within time points and in turn nested within participants), change in primary outcome measures (i.e., perceived stress and mood, salivary cortisol) will be analyzed by means of multilevel modeling using R packages (e.g., lme4 package [58]) to disentangle within-subject and between-subject variance. Level 1 comprises variables that are assessed repeatedly (i.e., the primary outcomes assessed before as well as immediately after each of the four clown visits), whereas level 2 specifies time points. Level 3 comprises of individual participants. Note that to prevent convergence issues, the best-fit model out of the following models will be used to test hypotheses: Fixed slope, random intercept for level 2 and 3 (Model 1); random slope at level 3 (Model 2); random slope at level 2 (Model 3); random slope at levels 2 and 3 (Model 4).

Given the complexity of the models, further analyses are to be considered exploratory in view of the small sample size of this pilot study. Potential moderator effects (number of visits, enjoyment, baseline variables) will be examined by testing cross-level interactions, provided the best-fit model allows for these analyses.

Missing data will be handled with full-information maximum likelihood (FIML) estimation, which uses all available information in the variance/covariance matrix to compute model parameters and produces less biased estimates than listwise or pairwise deletion or mean substitution [59]. Statistical significance will be assumed at $\alpha \leq .05$, as conventional. To reduce the chance of false discovery, alpha error correction will be applied using the Benjamini-Hochberg procedure [60].

## Ethics, data management and dissemination

### Research ethics approval

The present study has been reviewed and approved by the institutional review boards of the University of Vienna (Reference number: 00675; Date of approval: 3 May 2021; see S1 File) and the Medical University of Innsbruck (Reference number: 1272/2021; Date of approval: 24 November 2021). We comply with APA ethical standards and the Code of Ethics of the World Medical Association (Declaration of Helsinki) for research involving humans.

### Consent or assent

All clown artists, clinic staff and participants interested in participation will receive verbal and written information from the study staff. Prior to the start of data collection, written consent must be obtained from all participants and one legal guardian of each underaged participant, as well as an assent from the care staff.

### Confidentiality

We comply with statutory provisions concerning the collection, storage, and use of data in order to preserve full confidentiality (e.g., participants' and their legal guardians' written informed consent before participation). Data will be pseudonymized by a self-generated identification code. Documents with identifying information, such as signed consent forms, will be stored separately from the study data. Only authorized project staff members, who are subject to the obligation of secrecy, can identify an individual by linking a participant's name to the identification code.

### Data management

Collected data will be treated and stored in a pseudonymized form. All identifying information of participants will be removed from the datasets. Documents with identifying information, such as signed consent forms, will be stored securely and separately from the study data. One month after study completion all data are fully anonymized and hence contact details are irrevocably deleted. Data will be stored on a secured data server of the University of Vienna.

### Data monitoring

A data monitoring committee (DMC) is usually formed to monitor patient safety and treatment efficacy during an ongoing clinical trial. Since the intervention has no known risk and does not involve blinding, no DMC is formed. Furthermore, no interim analysis is conducted and no stopping guidelines are formulated.

### Harms

The intervention has a low risk of negative effects. Medical or paramedical care staff will independently evaluate the eligibility of current inpatients regarding potential negative impacts of clown visits or study participation on participants' health or well-being. Participants are only enrolled if there are no anticipated adverse consequences. During the intervention sessions participants are monitored by the care staff and examiners. If there are unexpected adverse effects, the present care staff takes care of the concerned participant. With regard to the measurements, no risk or complications are expected. All procedures used in the study are painless and harmless to health.

### Auditing

No auditing is planned.

### Access to data

Only authorized project staff members will have access to collected data, who are subject to the obligation of secrecy. Access to individual data is possible only in case of participants' study withdrawal for data deletion or if access is requested by participants or legal guardians.

### Dissemination policy

Study results will be made available to the scientific community and the cooperating health care facilities' staff. All reports of study results will be anonymous. Further, anonymized data and syntaxes of this study will be made openly available in OSF Storage (https://osf.io/) after completion of data collection.

## Discussion

To the best of our knowledge, the present pilot study will be the first to investigate the effects of interactive clown performances on children and adolescents in inpatient psychiatric care. Previous studies demonstrated that clown visits in healthcare settings reduce anxiety, stress, salivary cortisol, and pain of hospitalized children [15–20]. However, there is a lack of research on children and adolescents in the psychiatric setting who might especially be in need of and benefit from such clown visits. This pilot study aims at addressing this gap by examining the potential effects of clown interventions on stress reduction and mood improvement (primary outcomes) as well as longer-term changes in health-related quality of life, emotional and conduct problems, and perceived chronic stress (secondary outcomes). Since similar target variables have successfully been improved by healthcare clowning in pediatric patients in other clinical settings, it is important to explore whether children and adolescents in psychiatric treatment also benefit from such interventions. Scientific insights gained through this study can improve appropriate care in child and adolescent psychiatry, preventing negative outcomes typically elicited by inpatient stays, such as loneliness, loss of control, and other psychological impairments [6–8]. Findings will further provide knowledge about how clown visits influence patients' mental state and well-being from their own and the care staff's perspectives.

There are several strengths of this study. We will assess children's and adolescents' stress levels by subjective (self-report) as well as physiological measures (salivary cortisol), which enables us to measure the immediate effect on stress on two different levels, enhancing the validity of assessments. The inclusion of secondary outcomes provides further insights of the overall impact of the four-week intervention. As staff reports are collected as well, conclusions can be drawn from various perspectives. Experiences of care staff have only been included in a limited number of studies so far. More attention to this area is needed, as those few studies found positive effects of healthcare clowning on care staff and their working conditions in psychiatric settings [28, 31, 32]. Additionally, control and moderator variables are assessed. Thereby, we will be able to account for statistical controls as well as to identify potential subgroups benefiting most from interventions. Moreover, the repeated measures design (i.e., one clown visit on a weekly basis over four consecutive weeks) allows the investigation of cumulative effects as a result of repeated clown visits.

Although this pilot study aims at closing an important research gap, we are also aware of its limitations. First, the lack of an untreated control group will preclude causal inferences. Thus, there is a chance that alternative explanations such as other temporal developments could

account for found effects rather than the intervention per se. However, as assessments take place before and after each clown visit, within-subject temporal trends can still be examined. Second, due to the target sample size of approximately 50 children and adolescents and the circumstance that the study only takes place in two facilities, generalizability of study results is inherently limited. Third, not all questionnaires used in this study have been validated in previous research. We must therefore first examine whether the instruments in question, which were developed on the basis of theoretical considerations, also prove themselves empirically.

Nevertheless, depending on the outcomes and insights gained from this pilot study, future research including large-scale randomized controlled trials might prove worthy of investigation. Results will expand knowledge on the use of healthcare clowning to reduce stress and improve mood in children and adolescents in psychiatric care and might inform us about future clinical research within health-promoting measures in the context of child and adolescent psychiatry.

## Supporting information

**S1 Table. SPIRIT checklist.**
(DOC)

**S1 File. Ethics committee protocol (German original version and English translation).**
(PDF)

## Acknowledgments

We thank the two health care facilities and their staff members as well as the clown artists at RED NOSES Clowndoctors Austria that have agreed to cooperate with us to conduct this study.

## Author Contributions

**Conceptualization:** Martina Zemp, Urs M. Nater.

**Investigation:** Amos-Silvio Friedrich, Lorena Holzmeier.

**Methodology:** Martina Zemp, Urs M. Nater.

**Project administration:** Simone Seebacher.

**Resources:** Simone Seebacher, Urs M. Nater.

**Supervision:** Martina Zemp.

**Writing – original draft:** Martina Zemp, Amos-Silvio Friedrich, Lorena Holzmeier.

**Writing – review & editing:** Simone Seebacher, Maggie Rössler, Urs M. Nater.

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
