## [Decision Letter · Decision Letter 0]

11 Nov 2021

PONE-D-21-22963Effects of clown visits on stress and mood in children and adolescents in psychiatric care – Protocol for a pilot studyPLOS ONE

Dear Dr. Zemp,

Thank you for submitting your manuscript to PLOS ONE. After careful consideration, we feel that it has merit but does not fully meet PLOS ONE’s publication criteria as it currently stands. Therefore, we invite you to submit a revised version of the manuscript that addresses the points raised during the review process.

We look forward to receiving your revised manuscript.

Kind regards,

Ronald B. Gillam

Academic Editor

PLOS ONE

Journal Requirements:

2. Thank you for stating the following in the Competing Interests/Financial Disclosure* (delete as necessary) section:

“The authors have declared that no competing interests exist. Two authors (SS, MR) are employed at RED NOSES Clowndoctors Austria and International, respectively. These authors will not be involved in the clown visits, data collection procedures, or statistical analyses.”

We note that one or more of the authors are employed by a commercial company: RED NOSES Clowndoctors Austria and International

Additional Editor Comments (if provided):

Thank you for submitting your protocol to PLOS ONE. After reading your paper and the comments from the two reviewers, I find I am in agreement with Reviewer 2 who recommended a major revision. Please attend closely to Reviewer 2's comments and suggestions as you prepare your revision.

Sincerely,

Ronald Gillam, PhD.

Reviewers' comments:

Reviewer's Responses to Questions

**Comments to the Author**

1. Does the manuscript provide a valid rationale for the proposed study, with clearly identified and justified research questions?

Reviewer #1: Partly

Reviewer #2: Yes

2. Is the protocol technically sound and planned in a manner that will lead to a meaningful outcome and allow testing the stated hypotheses?

Reviewer #1: Partly

Reviewer #2: Yes

3. Is the methodology feasible and described in sufficient detail to allow the work to be replicable?

Reviewer #1: No

Reviewer #2: Yes

4. Have the authors described where all data underlying the findings will be made available when the study is complete?

Reviewer #1: Yes

Reviewer #2: Yes

5. Is the manuscript presented in an intelligible fashion and written in standard English?

Reviewer #1: Yes

Reviewer #2: Yes

6. Review Comments to the Author

You may also provide optional suggestions and comments to authors that they might find helpful in planning their study.

Reviewer #1: The manuscript entitled ‘Effects of clown visits on stress and mood in children and adolescents in psychiatric care – Protocol for a pilot study’ with the aims to investigate the short-term effects of clown visits by RED NOSES Clown doctors Austria on stress and mood levels of children and adolescents in psychiatric care.

The manuscript can be further improved based on the following comments.

Participants

Line 131, the term “self-reports” not clear whether it is referring to self-administered or the subject(s) providing the information to interviewer to be recorded. If it means own self filling in the information in the questionnaire, the word self-administered would be preferred.

Line 147, which effect size figure was used in the calculation, 1 or 2-tailed test and whether the effect size was based on children or adolescent or both to derive the sample size 30 to be clearly stated.

Trial design

For the questionnaire’s administration for the children, adolescents and care staff, the word self-administered or interviewed to be clearly stated. Information on who will be administering the questionnaires, involved in the saliva samples collection to be clearly stated.

Primary outcome measures

Line 195, the statement unclear and more information to be provided.

Line 198, the German version of MDMQ to be stated.

Evaluation of care staff

Line 249, information on whether the items for self-developed questionnaire were fully adopted from the prior existing tools and no changes or revision or no new items construction required to develop the questionnaire to be clearly stated.

Sociodemographic and clinical data and control variables

Line 268, 279, 283, how/who is administering the questionnaire to the subjects and who capturing the information to be stated.

Line 273-277, how or who capturing this data to be clearly stated.

Line 289-291, how the questionnaire will be administered to be clearly stated.

Statistical analyses

The level of the accepted statistical significance to be stated.

Since this is a pilot study and the design is not involving control group and the sample size is based on the current design, thus comprehensive statistical analyses to be avoided.

Research ethics approval

Line 322-326, statements were repeated as in Line 170-174.

Discussion

Line 411, if the purpose of this study is not to include the control group, therefore this limitation not to be mentioned. Likewise, with sample size limitation, Line 415-417.

If there are hypotheses and hypotheses proving to be done, control group for comparison would be useful (e.g without any intervention involved).

Not all references conformed to the journal format.

Reviewer #2: Considering the existence of the evidence, why should the authors carry out a pilot study. Instead, it would be better to do a full-blown main study.

7. PLOS authors have the option to publish the peer review history of their article (what does this mean?). If published, this will include your full peer review and any attached files.

Reviewer #1: No

Reviewer #2: No

---

## [Author Response · Author response to Decision Letter 0]

7 Dec 2021

Dear Editor, 

Dear Reviewers, 

We would like to thank the editor and the two anonymous reviewers again for the helpful comments about our manuscript. We responded to each point and made changes to the manuscript (see file labeled 'Revised Manuscript with Track Changes'). As requested, we also submit an unmarked version of the manuscript without tracked changes (see file labeled 'Manuscript'). The line numbers we provided in this response letter refer to the first, marked version of the manuscript. 

To comprehensively respond to all reviewer feedback, we pasted their comments into this document. Our responses are noted with Author Response in bold type and we note the location (lines) of any changes made to the manuscript. 

Once again, thank you very much for the time and attention you devoted to this manuscript. We believe these revisions resulted in a better quality piece of work.

Sincerely, 

Authors

 

Reviewer #1: 

Comment 1: Line 131, the term “self-reports” not clear whether it is referring to self-administered or the subject(s) providing the information to interviewer to be recorded. If it means own self filling in the information in the questionnaire, the word self-administered would be preferred.

Author Response: Thank you for bringing this imprecision to our attention. We replaced “self-reports” by “self-administered questionnaires”, where applicable (see lines 131; 147; 167; 169; 171 etc.). Additionally, we added a general note about the kind of administration to the procedure section (see lines 164-166): “Participants will be administered paper-pencil versions of the undermentioned measures and will be asked to fill them out by themselves under supervision of the examiners.” Thus, the questionnaires were self-administered (no interview or recorded data).

Comment 2: Line 147, which effect size figure was used in the calculation, 1 or 2-tailed test and whether the effect size was based on children or adolescent or both to derive the sample size 30 to be clearly stated.

Author Response: Thank you for pointing to this lack of information. We revised this paragraph accordingly: We added the effect size used in the calculations for salivary cortisol and clarified the distinction from self-administered measures (line 147). Samples of reference studies were children for all except of emotional state estimates (lines 149-151). Our calculations revealed that a minimum sample size of about 30 participants in total (children and adolescents together) is needed (line 148). One-tailed testing was specified (line 146).

Comment 3: For the questionnaire’s administration for the children, adolescents and care staff, the word self-administered or interviewed to be clearly stated. Information on who will be administering the questionnaires, involved in the saliva samples collection to be clearly stated.

Author Response: Please see our answer to your Comment 1. We specified self-administered questionnaires where applicable throughout the manuscript. Concerning the saliva samples collection, we tried to describe the procedure more clearly (see lines 222-226): “Samples will be collected immediately before as well as immediately after each of the four clown visits in a self-administered way, but under the supervision of the examiners. Participants will be thoroughly instructed to collect accumulated saliva for two minutes without swallowing. After the two minutes, participants will transfer the accumulated saliva into the Salicap tube via the straw.”

Comment 4: Line 195, the statement unclear and more information to be provided.

Author Response: We provided better information about the assessment of the perceived stress levels of children and adolescents, and also cited previous research that has used the VAS measurement approach (lines 200-207).

Comment 5: Line 198, the German version of MDMQ to be stated.

Author Response: The MDMQ is the English version of an originally German instrument. To alleviate confusion, we now state the use of the original German version (line 209).

Comment 6: Line 249, information on whether the items for self-developed questionnaire were fully adopted from the prior existing tools and no changes or revision or no new items construction required to develop the questionnaire to be clearly stated.

Author Response: We agree that this was not clearly stated in the previous submission. We revised the description of this measure in lines 267-271: “The questionnaire was developed by the authors specifically for the study purpose on the basis of previously implemented research tools [32,55]. The items were partly self-developed and partly translated into German, adopted according to the study’s objectives, newly designed and formulated in accordance with previous research [32,55].”

As this is a caveat of our study, we added a limitation to the discussion (lines 457-460): “Third, not all questionnaires used in this study have been validated in previous research. We must therefore first examine whether the instruments in question, which were developed on the basis of theoretical considerations, also prove themselves empirically.”

Comment 7: Line 268, 279, 283, how/who is administering the questionnaire to the subjects and who capturing the information to be stated.

Author Response: Please see Comment 1; we specified self-administered questionnaires where applicable.

Comment 8: Line 273-277, how or who capturing this data to be clearly stated.

Author Response: We revised the entire measures section to make clear who is reporting what data (see lines 200; 223-224; 294; etc.).

Comment 9: Line 289-291, how the questionnaire will be administered to be clearly stated.

Author Response: Please see Comment 1; we specified self-administered questionnaires where applicable.

Comment 10: The level of the accepted statistical significance to be stated.

Author Response: We added the note about accepted statistical significance in line 350: “Statistical significance will be assumed at α ≤ .05, as conventional.” We further added a method of alpha correction (lines 350-352): “To reduce the chance of false discovery, alpha error correction will be applied using the Benjamini-Hochberg procedure [60].”

Comment 11: Since this is a pilot study and the design is not involving control group and the sample size is based on the current design, thus comprehensive statistical analyses to be avoided.

Author Response: The proposed statistical model, while complex, is needed to analyze our nested data appropriately. It allows us to examine all instances of pre-post data across the repeated clown visits, resulting in a larger number of data points (up to four clown visits per person). Hence, sample size of 30-50 participants with repeated measurement brings higher statistical power. Higher levels of the model are needed to disentangle variance components between participants, time points, and research sites as not to overinterpret potential effects resulting from factors other than the intervention. Nevertheless, your comment serves as an important reminder to keep the scope of the analyses on a level with the aims of this pilot study. We added the note that the more complex analyses are to be considered exploratory given the rather small sample size and revised the entire section about the statistical analysis according your comment (see lines 343-344).

Comment 12: Line 322-326, statements were repeated as in Line 170-174.

Author Response: Thank you for noticing this redundancy. The first occurrence of IRB statements has been deleted so that ethics approval is only mentioned once in the revised manuscript (see lines 354-362).

Comment 13: Line 411, if the purpose of this study is not to include the control group, therefore this limitation not to be mentioned. Likewise, with sample size limitation, Line 415-417.

Author Response: We acknowledge that these limitations can be deemed redundant under the premise of our pilot study design. As elaborated on in our response to Comment 14, we are aware of the limited validity associated with our non-controlled single-group design. We included these sentences as a reminder for cautious interpretation of any results. For this reason, we would prefer to keep this explicit statement in the interest of an informed discussion, but slightly revised the paragraph (specifically, the mention of the limited statistical power was removed given our power calculations; see line 456). We hope that the Reviewer understands this decision.

Comment 14: If there are hypotheses and hypotheses proving to be done, control group for comparison would be useful (e.g without any intervention involved).

Author Response: We recognize the importance of controlled trials in treatment research and are planning to involve a control group in future studies. For this pilot study, we opted for a simple non-controlled pre-/post-test design (please see also our response to Reviewer #2). While we agree that a control group would be needed in causally relating potential changes to the intervention, our design does allow for a comparison of pre-post differences under the premise that only the intervention took place between the two time points. Our stated hypotheses at the end of the introduction (see lines 114-121) are formulated appropriately, that is, in a way that they can be tested without a control group.

Comment 15: Not all references conformed to the journal format.

Author Response: We appreciate your thorough examination. After double-checking reference formatting, corrections were made where necessary.

Reviewer #2:

Considering the existence of the evidence, why should the authors carry out a pilot study. Instead, it would be better to do a full-blown main study.

Author Response: Thanks for raising a valid point in suggesting an immediate main study of larger scope on the basis of promising evidence. However, our decision to prepend a slim pilot study is based on the following arguments: 

As elaborated in the background section, even though there is some promising evidence on the effectiveness of clown visits in the general pediatric setting as well as adult psychiatry, there is currently a complete lack of pertinent studies in child and adolescent psychiatry. Therefore, we consider the state of the evidence base to be too little advanced for a full-blown main study. In this vein, we attempt to pave the road for such comprehensive main studies with two goals in mind: Firstly, we aim at gaining preliminary data on the (as of now, assumed) effectiveness in this target population in order to evaluate if future RCTs are worth pursuing, to identify effect sizes that are to be expected, and to determine sample sizes needed. The second main goal of this pilot study is to gather insight into the feasibility of such studies due to circumstances pertaining to the complex setting. For example, as described in the manuscript, type of setting (group or individual), duration of clown visits, and artistic sequences can vary across visits, instead of being standardized. The specific artistic sequences and type of setting are implemented spontaneously according to the situational atmosphere and current condition of participants. One clown visit in the group setting will last between 1 and 2 hours including individual as well as collective attention depending on age, number of participants, and situational atmosphere. In planning and cooperating with the participating institutions, we have already been able to gain important experience in implementing this time-sensitive research paradigm (including questionnaires fulfilled by children and adolescents as well as care staff, collection of saliva samples for assessing salivary cortisol prior to and after clown visits etc.) into clinical and artistic routines. We hope that we are able to successfully transfer these learnings to a larger study using a well thought-out trial design including randomization and control groups in the future.

A further argument for the pilot study were and still are the circumstances surrounding the current global pandemic that, as cooperating hospital staff have repeatedly expressed, have placed an additional load on care routines. It was thus also a reason to start with a pilot study that minimally impacts these procedures.

We hope that we have provided an understandable line of reasoning for our decision in favor of a pilot study prior to a full-blown main study.

---

## [Decision Letter · Decision Letter 1]

2 Feb 2022

Effects of clown visits on stress and mood in children and adolescents in psychiatric care – Protocol for a pilot study

PONE-D-21-22963R1

Dear Dr. Zemp,

We’re pleased to inform you that your manuscript has been judged scientifically suitable for publication and will be formally accepted for publication once it meets all outstanding technical requirements.

Kind regards,

Ronald B. Gillam

Academic Editor

PLOS ONE

Additional Editor Comments (optional):

Thank you for sending your revision of PONE-D-21-22963R1. I agree with both reviewers that you addressed their initial questions and suggestions successfully. Therefore, I recommend acceptance of the revised paper.

Reviewers' comments:

Reviewer's Responses to Questions

**Comments to the Author**

1. Does the manuscript provide a valid rationale for the proposed study, with clearly identified and justified research questions?

Reviewer #1: Partly

Reviewer #2: Yes

2. Is the protocol technically sound and planned in a manner that will lead to a meaningful outcome and allow testing the stated hypotheses?

Reviewer #1: Partly

Reviewer #2: Yes

3. Is the methodology feasible and described in sufficient detail to allow the work to be replicable?

Reviewer #1: Yes

Reviewer #2: Yes

4. Have the authors described where all data underlying the findings will be made available when the study is complete?

Reviewer #1: Yes

Reviewer #2: Yes

5. Is the manuscript presented in an intelligible fashion and written in standard English?

Reviewer #1: Yes

Reviewer #2: Yes

6. Review Comments to the Author

You may also provide optional suggestions and comments to authors that they might find helpful in planning their study.

Reviewer #1: The authors have addressed the comments. I have no further comments. The manuscript is acceptable for publication.

Reviewer #2: Nil. The queries are answered satisfactorily...................................................................................................................................

7. PLOS authors have the option to publish the peer review history of their article (what does this mean?). If published, this will include your full peer review and any attached files.

Reviewer #1: No

Reviewer #2: No

---

## [Editor Report · Acceptance letter]

8 Feb 2022

PONE-D-21-22963R1 

Effects of clown visits on stress and mood in children and adolescents in psychiatric care – Protocol for a pilot study 

Dear Dr. Zemp:

I'm pleased to inform you that your manuscript has been deemed suitable for publication in PLOS ONE. Congratulations! Your manuscript is now with our production department. 

Kind regards, 

on behalf of

Dr. Ronald B. Gillam 

Academic Editor

PLOS ONE